# Crosstalk Pathway between Trehalose Metabolism and Cytokinin Degradation for the Determination of the Number of Berries per Bunch in Grapes

**DOI:** 10.3390/cells9112378

**Published:** 2020-10-29

**Authors:** Ayane Moriyama, Chiho Yamaguchi, Shinichi Enoki, Yoshinao Aoki, Shunji Suzuki

**Affiliations:** Laboratory of Fruit Genetic Engineering, The Institute of Enology and Viticulture, University of Yamanashi, 1-13-1 Kofu, Yamanashi 400-0005, Japan; g20lf007@yamanashi.ac.jp (A.M.); wmika@yamanashi.ac.jp (C.Y.); senoki@yamanashi.ac.jp (S.E.); yaoki@yamanashi.ac.jp (Y.A.)

**Keywords:** berry number, cytokinin oxidase/dehydrogenase, grapevine, inflorescence, sister of ramosa3, trehalose

## Abstract

In grapes, the number of flowers per inflorescence determines the compactness of grape bunches. Grape cultivars with tight bunches and thin-skinned berries easily undergo berry splitting, especially in growing areas with heavy rainfall during the grapevine growing season, such as Japan. We report herein that grape cytokinin oxidase/dehydrogenase 5 (VvCKX5) determines the number of berries per inflorescence in grapes. The number of berries per bunch was inversely proportional to the *VvCKX5* expression level in juvenile inflorescences among the cultivars tested. VvCKX5 overexpression drastically decreased the number of flower buds per inflorescence in *Arabidopsis* plants, suggesting that VvCKX5 might be one of the negative regulators of the number of flowers per inflorescence in grapes. Similarly, the overexpression of grape sister of ramose 3 (VvSRA), which encodes trehalose 6-phosphate phosphatase that catalyzes the conversion of trehalose-6-phosphate into trehalose, upregulated *AtCKX7* expression in *Arabidopsis* plants, leading to a decrease in the number of flower buds per *Arabidopsis* inflorescence. *VvCKX5* gene expression was upregulated in grapevine cultured cells and juvenile grape inflorescences treated with trehalose. Finally, injecting trehalose into swelling buds nearing bud break using a microsyringe decreased the number of berries per bunch by half. VvCKX5 overexpression in *Arabidopsis* plants had no effect on the number of secondary inflorescences from the main inflorescence, and similarly trehalose did not affect pedicel branching on grapevine inflorescences, suggesting that VvCKX5, as well as VvSRA-mediated trehalose metabolism, regulates flower formation but not inflorescence branching. These findings may provide new information on the crosstalk between VvSRA-mediated trehalose metabolism and VvCKX-mediated cytokinin degradation for determining the number of berries per bunch. Furthermore, this study is expected to contribute to the development of innovative cultivation techniques for loosening tight bunches.

## 1. Introduction

The size and density of grape bunches affect wine quality. Tight bunches of small and thin-skinned Pinot Noir (*Vitis vinifera* L.) berries easily undergo berry splitting [1], which in turn leads to phytopathogenic disease and insect damage. In Japan, in particular, heavy rainfall before véraison (from the beginning of June to the middle of July) results in the swelling of Pinot Noir berries, producing cracks on the berries’ skin. Spraying *V. vinifera* grape bunches with gibberellin was found to lead to elongation of the inflorescence peduncle, thereby reducing the density of bunches [2,3,4]. However, gibberellin application has not been carried out in viticulture for wine grapes. There are quite a few vineyard operators who conduct berry thinning to loosen Pinot Noir bunches in Japan; however, this is not viable because the average age of operators is advancing and the number of operators is declining in Japan.

Grape bunch compactness is determined by berry number, berry trait, and rachis trait [5]. Environmental conditions, including temperature and rainfall [6,7], affect berry number as well as reproductive performance, including fruit set rate [8]. Quantitative trait locus analysis demonstrated that only one single nucleotide polymorphism in the myeloblastosis viral oncogene homolog (MYB)-type transcription factor MYB108B was associated with berry number in two seasons out of three consecutive seasons, suggesting a high sensitivity of berry number to seasonal environmental changes [9].

Axillary meristems produce either branches or flowers on plant inflorescences [10]. ramosa, identified in maize, regulates the fate of most axillary meristems during inflorescence development [10,11,12]. Ramosa1 (RA1) and ramosa2 are transcription factors that affect the early development of maize inflorescences. RAMOSA3 and sister of ramosa3 (SRA) encode trehalose 6-phosphate phosphatases [11]. Trehalose metabolism is thought to function as a transcriptional regulator at the upstream of RA1 during inflorescence development. In grapevines, the transcription level of *V. vinifera* SRA, VvSRA, is negatively correlated with rachis development on the lateral branch of *V. vinifera* inflorescences, thereby decreasing the number of berries per bunch in cultivars expressing VvSRA abundantly [13]. Concomitantly, the overexpression of VvSRA in *Arabidopsis* plants drastically decreased the number of flower buds on secondary inflorescences [13]. In addition, VvSRA overexpression induced smaller flowers, shorter inflorescences, and expanded rosette leaves in *Arabidopsis* plants [13]. Ramosa3-mediated trehalose metabolism was shown to control inflorescence branching by modification of a sugar signal into axillary meristems [11]. Although the molecular mechanism underlying the number of flowers per inflorescence in grapes has not been established, VvSRA-mediated trehalose metabolism might be one of the key regulators during inflorescence development in *V. vinifera* grapes.

Trehalose confers tolerance to desiccation from drought, salt, and/or low-temperature stresses on plants [14]. Transgenic rice with trehalose-6-phosphate synthase/phosphatase accumulated 3−10 times more trehalose than control rice, resulting in high tolerance to salt and drought stresses [15]. Foliar application of trehalose improved tolerance to drought stress in maize through improvement of photosynthetic attributes and water retention [16]. Trehalose is also involved in plant growth. *Arabidopsis* seedlings grown on trehalose stopped growing due to rapid accumulation of trehalose-6-phosphate [17], whereas foliar application of trehalose at the vegetative growth stage increased fresh and dry weights of shoots in maize [16]. However, so far, the function of trehalose in flower development has remained a mystery.

In the present study, we propose the hypothesis that VvSRA-mediated trehalose metabolism activates cytokinin oxidase/dehydrogenase (CKX)-mediated cytokinin degradation in meristems of inflorescences, thereby decreasing the number of berries per bunch. CKX catalyzes irreversible cytokinin degradation [18]. Cytokinin promotes floral transition in *Arabidopsis* plants [19]. In grapevines, cytokinin is implicated in the control of inflorescence formation, differentiation of flowers, pistil development, and berry development [20]. However, the role of cytokinin in the control of the number of berries per bunch for each cultivar is not well understood. In rice, the reduction of cytokinin oxidase/dehydrogenase (*OsCKX2*) gene expression causes cytokinin accumulation in meristems of inflorescences and results in an increase in grain number in the main panicle [21]. Herein, we demonstrate that (1) VvSRA overexpression upregulated cytokinin oxidase/dehydrogenase gene expression in *Arabidopsis* plants, (2) trehalose upregulated *VvCKX5* gene expression in grapevine cultured cells and juvenile grape inflorescences, (3) there was a strong negative correlation between *VvCKX5* gene expression level in juvenile inflorescences and the number of berries per bunch among *V. vinifera* cultivars, and (4) VvCKX5 overexpression drastically decreased the number of flower buds per inflorescence in *Arabidopsis* plants. Finally, we succeeded in decreasing the number of berries per bunch in field-grown Pinot Noir by injecting trehalose into swelling buds nearing bud break using a microsyringe in the 2018 and 2020 growing seasons.

## 2. Materials and Methods

### 2.1. Plant Materials

Grapevines (*Vitis vinifera* cvs. Pinot Noir, Chardonnay, Riesling, and Koshu), grafted onto Kober 5BB, were cultivated in the experimental vineyard of The Institute of Enology and Viticulture, University of Yamanashi, Yamanashi, Japan. The grapevines were approximately 30 years old and trained in double cordon style.

Grapevine cultured cells prepared from meristems of *V. vinifera* cv. Koshu were maintained on modified Gamborg’s B5 medium at 27 °C in the dark [22]. VvSRA-overexpressing *Arabidopsis thaliana* plants were prepared in our previous study [13].

### 2.2. Number of Berries per Bunch

Ten bunches were collected from each cultivar on 5 August 2014 and 2 August 2018 (each 10 weeks after flowering), respectively. The number of berries per bunch was counted.

### 2.3. Trehalose Treatment of Grapevine Cultured Cells

Grapevine cultured cells were incubated on modified Gamborg’s B5 medium supplemented with 0.5%, 1%, or 5% trehalose at 27 °C in the dark for 6 h. As control experiments, grapevine cultured cells were incubated on modified Gamborg’s B5 medium supplemented with 5% maltose or cellobiose at 27 °C in the dark for 6 h. After incubation, the cells were frozen immediately in liquid nitrogen and subjected to RNA isolation.

### 2.4. Trehalose Treatment of Juvenile Grape Inflorescences

Five juvenile grape inflorescences at Eichhorn–Lorenz Stage 12 (five to six leaves unfolded; approximately 5−8 mm inflorescences clearly visible) were treated with 200 μL of 10% trehalose containing 0.1% Approach BI (Kao Chemicals, Osaka, Japan) on 23 April 2018. At 0, 6, or 24 h post treatment, the inflorescences were collected and frozen immediately in liquid nitrogen.

### 2.5. NAA and Kinetin Treatment of Grapevine Cultured Cells

Grapevine cultured cells were incubated on modified Gamborg’s B5 medium supplemented with 5 mM α-naphthaleneacetic acid (NAA) or 2.5 mg/mL kinetin (five times the normal addition, respectively) at 27 °C in the dark. At 0, 6, or 24 h post treatment, the cells were frozen immediately in liquid nitrogen and subjected to RNA isolation.

### 2.6. RNA Isolation

Grapevine cultured cells frozen in liquid nitrogen were homogenized using an SK mill (SK-200, Tokken, Kashiwa, Japan). Juvenile grape inflorescences were placed in a mortar containing liquid nitrogen and homogenized with a pestle. Total RNA was isolated from the pulverized samples using RNAiso Plus (Takara Bio, Otsu, Japan) and then purified with a NucleoSpin RNA Plant (Takara Bio) according to the manufacturer’s instructions.

### 2.7. Real-Time RT-PCR

cDNA was synthesized from total RNA using a Prime Script RT Reagent Kit with gDNA Eraser (Takara Bio) according to the manufacturer’s instructions. Real-time RT-PCR was performed using a Thermal Cycler Dice Real Time System (Takara Bio) with TB Green Premix Ex Taq II (Takara Bio). PCR amplification was performed for 40 cycles at 95 °C for 5 s, and at 60 °C for 30 s after an initial denaturation at 95 °C for 30 s. The nucleotide sequences of the PCR primers used in the present study are as follows: VvCKX1-like primers (5′-CAACAGACACCATTACCTTCCATC-3′ and 5′-GCCTGTGCTTGACCTTGGA-3′, GenBank accession no. XM_002284524), VvCKX3 primers (5′-CTGGTAAAGGGGAACTTGTGACTT-3′ and 5′-TCGGGTATTCAGAGGGTGAGA-3′, GenBank accession no. XM_002264409), VvCKX5 primers (5′-ACCTTAGACCCCACCGACCT-3′ and 5′-TCACACCCTTTGACGCACTC-3′, GenBank accession no. XM_002280761), VvCKX7 primers (5′-GTCTGTATGCTGTGAAGCCTGTG-3′ and 5′-ACCACCTCAATCCGCTCCT-3′, GenBank accession no. XM_002279924), VvCKX9 primers (5′-GGGCTTCTGTCTTTGGCTTC-3′ and 5′-AACTGAAATGCCCGTCAACA-3′, GenBank accession no. XM_002270805), grape actin primers (5′-CAAGAGCTGGAAACTGCAAAGA-3′ and 5′-AATGAGAGATGGCTGGAAGAGG-3′, GenBank accession no. AF369524), *Arabidopsis* cytokinin oxidase/hydrogenase 7 (AtCKX7) primers (5′-CGTTCGATTACGTGGAAGGAT-3′ and 5′-ACCCGACTACCACAATCTTG-3′, GenBank accession no. AF303981), and *Arabidopsis* actin primers (5′-AACTCCATAATGAAGTGTGA-3′ and 5′-TTGAACTCAGAAGCACTTCC-3′, GenBank accession no. NM_179953). Raw expression data were analyzed using Thermal Cycler Dice Real Time System Single Software ver. 3.00 (Takara Bio). Actin was used as the internal control for normalization. To verify the specificity of the amplification reaction, the dissociation curves for each sample were analyzed. Expression levels were determined as the number of amplification cycles needed to reach a fixed threshold using the standard curve method. Finally, the expression level of each gene was expressed as a relative value of the actin internal control.

### 2.8. Simple Linear Regression Analysis

To evaluate the correlation between the number of berries per bunch and *VvCKX* expression levels in juvenile grape inflorescences, simple linear regression analysis was performed using Excel statistics software 2012 (Social Survey Research Information, Tokyo, Japan). The number of berries per bunch for each cultivar was used as the dependent variable. The relative expression level of each gene in the juvenile grape inflorescences of each cultivar was used as the explanatory variable.

### 2.9. Overexpression of VvCKX5 in Arabidopsis Plants

The open reading frame of *VvCKX5* was synthesized by amplification of total RNA by RT-PCR. The nucleotide sequences of the primers were as follows: 5′-GGTACCATGGCGCCGAAGCTTCT GCT-3′ containing a *Kpn*I site (underlined) and 5′-GAATTCTCACCAAGAAGAAGCCGCCA-3′ containing an *Eco*RI site (underlined). The PCR product was digested with *Kpn*I and *Eco*RI and ligated to the *Kpn*I and *Eco*RI sites of the binary vector pRI101-AN (Takara), resulting in a VvCKX5 expression plasmid. *A. thaliana* Col-0 was transformed by the floral dip method through *Agrobacterium tumefaciens* strain LBA4404 with the plasmid [23]. We obtained three independent T4 homozygous transformants denoted by VvCKX5-OE3, VvCKX5-OE4, and VvCKX5-OE6. The expression levels of the *VvCKX5* transgene in each transformant were measured by real-time RT-PCR.

### 2.10. Phenotypic Analysis of VvCKX5-Overexpressing Arabidopsis Plants

T4 seeds were sown on rockwool blocks (2.5 cm × 2.5 cm × 3.8 cm) and placed in an incubator set at 22 °C (11.8 Wm-2/16 h/d). The seedlings were planted in soil and grown in the same incubator. Different phenotypes of VvCKX5-overexpressing plants were photographed at each growth stage and compared with those of wild type plants. Plant height was measured on day 44 (wild type) or 57 (transgenic) after sowing. The numbers of secondary inflorescences and flower buds on plants were counted on day 42 (wild type) or 57 (transgenic) after sowing. Flower size was measured on day 35 (wild type) or 52 (transgenic) after sowing. Silique length was measured on day 42 (wild type) or 69 (transgenic) after sowing.

### 2.11. Trehalose Treatment of Buds of Field-Grown Grapevines

Grapevines (*V. vinifera* cv. Pinot Noir) cultivated in the experimental vineyard of The Institute of Enology and Viticulture, University of Yamanashi, Yamanashi, Japan, were used. Ten buds that were nearing bud break (Eichhorn–Lorenz Stage 3, brownish wool clearly visible, Appendix A) were injected with 10% trehalose or water containing 0.1% Approach BI (250 μL/bud) using a microsyringe (Ito, Shixuoka, Japan, Appendix A) in the 2018 and 2020 growing seasons (4 April 2018 and 8 April 2020, respectively). As a control, buds were sprayed with 10% trehalose or water containing 0.1% Approach BI (250 μL/bud) using an atomizer. Young shoots (Eichhorn–Lorenz Stages 5–7, bud burst or first leaf unfolded) from the buds were again sprayed with 10% trehalose or water containing 0.1% Approach BI (250 μL/shoot) using an atomizer 10 days post first treatment (Appendix A). Bunches on each shoot were collected at harvest (Eichhorn–Lorenz Stage 38 on 2 August 2018 and 9 August 2020, respectively) and subjected to analysis of bunch and berry characteristics.

### 2.12. Bunch and Berry Characteristics

Twenty bunches were used for the measurement of berry characteristics. Weight and vertical length of each bunch were measured using an electronic balance and a digital caliper (EK2000i, A & D Co., Tokyo, Japan), respectively. The numbers of berries per bunch and the number of pedicels per inflorescence were counted. Ten berries (two from the top of the bunch, six from the middle of the bunch, and two from the bottom of the bunch) were collected from each bunch, and the weights of the ten berries were measured using an electronic balance.

Juices were obtained from the whole bunch after the ten berries had been removed by hand-pressing. Soluble solids content (Brix) and total acids content (g/100 mL) in the juices were measured with a refractometer (PAL-BX/ACID2, Atago, Tokyo, Japan). Skins of the ten berries were peeled off with tweezers and frozen immediately in liquid nitrogen. Extraction of anthocyanins from berry skins and measurement of total anthocyanin content were performed as described previously [24] with some modification. Briefly, one gram of the pulverized sample was macerated in 10 mL of HCl-methanol (36:1 (*v*/*v*)) at room temperature in the dark overnight. After mixing, the absorbance (OD_520_) of the mixture was measured using a spectrometer (UV-1800, Shimadzu, Kyoto, Japan). Total anthocyanin content was calculated by using a previously published formula [25] and converted into malvidin-3-glucoside equivalent as mg per gram of fresh berry skin weight.

### 2.13. Statistical Analysis

Data are presented as means ± standard deviations. Statistical analysis was performed by Student’s *t*-test or Dunnett’s test using Excel statistics software 2012.

## 3. Results

### 3.1. Number of Berries per Bunch Differs among Grape Cultivars

Four cultivars showing different phenotypes of grape inflorescences (Figure 1) were cultivated in the 2014 and 2018 growing seasons using the same practice and training system in the experimental vineyard of The Institute of Enology and Viticulture, University of Yamanashi. Grape inflorescences were composed of lateral branches and berries formed on the pedicels that developed on the branches. The number of berries per bunch differed among the cultivars tested (Figure 1). Pinot Noir had more than one hundred berries on a bunch, whereas Koshu, an indigenous Japanese *Vitis* species [26], showed the lowest number of berries (approximately 30 berries per bunch) among the cultivars tested under our cultivation conditions.

### 3.2. VvSRA Overexpression Increases Cytokinin Oxidase/Hydrogenase Expression in Arabidopsis Plants

The overexpression of VvSRA in *Arabidopsis* plants drastically decreased the number of flower buds per inflorescence [13]. In VvSRA-overexpressing *Arabidopsis* plants, the expression of *AtCKX7*, an *A. thaliana* homolog of *OsCKX2*, was drastically upregulated compared with wild type (Figure 2). This suggests that VvSRA-mediated trehalose metabolism promotes cytokinin oxidase/hydrogenase expression, thereby suppressing flower development.

### 3.3. VvCKX5 Gene Expression Is Negatively Correlated with Number of Berries per Bunch

*VvCKX1-like*, *VvCKX3*, *VvCKX5*, *VvCKX7*, and *VvCKX9* were selected from the GenBank database. The relationship between the number of berries per bunch and *VvCKX* expression in juvenile grape inflorescences was investigated. *VvCKX3* was not expressed in juvenile inflorescences irrespective of the cultivar. *VvCKX1-like*, *VvCKX5*, *VvCKX7*, and *VvCKX9* were expressed in the juvenile inflorescences (Figure 3A). *VvCKX5*, *VvCKX7*, and *VvCKX9* expression was the lowest in the juvenile inflorescences of Pinot Noir with the highest number of berries per bunch, whereas these transcripts were most abundantly expressed in Koshu with the lowest number of berries per bunch. *VvCKX1-like* expression in Pinot Noir and Koshu was similar.

The regression lines demonstrated a negative correlation between the number of berries per bunch and *VvCKX5*, *VvCKX7*, or *VvCKX9* expression in the juvenile inflorescences (Figure 3B). *VvCKX1-like* expression was not correlated with the number of berries per bunch. Simple linear regression analysis demonstrated that *VvCKX5* expression in the juvenile inflorescences showed a strong negative correlation with the number of berries per bunch (*p* = 0.0023, Table 1). *VvCKX7* and *VvCKX9* expression in the juvenile inflorescences had no effect on the number of berries per bunch.

Taken together, these results suggest that *VvCKX5* expression is negatively correlated with the number of berries per bunch.

### 3.4. Trehalose Upregulates VvCKX5 Gene Expression in Grapevine Cultured Cells and Juvenile Grape Inflorescences

To determine whether exogenous trehalose upregulates *VvCKX5* gene expression in grapevines, VR cells were used for endogenous trehalose treatment (Figure 4). VR cells grown on medium supplemented with 5% trehalose had significantly higher *VvCKX5* gene expression than the control at 6 h post treatment (Figure 4A). Low concentrations of trehalose (0.5% and 1%) did not induce *VvCKX5* gene expression in VR cells (Figure 4B). Disaccharides, including cellobiose and maltose, had no effect on *VvCKX5* gene expression in VR cells (Figure 4C), suggesting that osmotic stress to grapevine cultured cells by disaccharides was independent of upregulation of *VvCKX5* gene expression by trehalose.

Juvenile inflorescences of field-grown grapevines were treated with trehalose (Figure 4D). Exogenous trehalose upregulated *VvCKX5* gene expression in the juvenile inflorescences at 6 h and 24 h post treatment.

Taken together, these results suggest that exogenous trehalose, but not other disaccharides, upregulates *VvCKX5* gene expression in grapevines.

### 3.5. NAA Upregulates VvCKX5 Gene Expression in Grapevine Cultured Cells

VR cells grown on medium supplemented with an excess amount of NAA had significantly higher *VvCKX5* gene expression than the control at 6 and 24 h post treatment (Figure 5). An excess amount of kinetin did not induce *VvCKX5* gene expression in VR cells.

### 3.6. VvCKX5 Overexpression Has a Notable Effect on Flower Number in Arabidopsis Plants

We created transgenic *Arabidopsis* plants overexpressing *VvCKX5* to evaluate whether VvCKX5 affects the number of flowers per inflorescence. Three homozygous transgenic lines, VvCKX5-OE3, VvCKX5-OE4, and VvCKX5-OE6, were obtained. All lines constitutively overexpressed *VvCKX5* transcripts in rosette leaves of 36-day-old plants (Figure 6A).

The germination rates and times of T4 seeds of transgenic lines were comparable to those of wild type (data not shown). The phenotypes of the transgenic plants were different from those of wild type on day 44 after sowing (Figure 6B). Wild type showed secondary inflorescences and flower bud formation on day 44 after sowing (Figure 6B), whereas only the main inflorescence emerged in the overexpression lines (Figure 6B,C). On day 57 after sowing, the overexpression lines showed secondary inflorescences and flower bud formation (Figure 6C). The heights of the transgenic plants were approximately half to two-thirds of that of wild type (Figure 6D). The numbers of secondary inflorescences on the transgenic plants were comparable to that of wild type on day 57 after sowing (Figure 6E), suggesting no effect of VvCKX5 overexpression on inflorescence branching.

The number of flower buds on the inflorescences of the transgenic plants was less than one-third to half of that of wild type (Figure 7A). Phenotypical differences in flowers and siliques were observed between the transgenic plants and wild type (Figure 7B,D). Flowers of the transgenic plants were significantly smaller than that of wild type (Figure 7C). Consequently, the siliques of the transgenic plants were shorter than that of wild type (Figure 7E).

### 3.7. Trehalose Injection into Buds Decreases the Number of Berries per Bunch

To evaluate whether exogenous trehalose application to field-grown grapevines affects the number of berries per bunch, we performed the injection of trehalose into swelling buds nearing bud break (Eichhorn–Lorenz Stage 3) using a microsyringe (Supplemental Appendix A) in the 2018 and 2020 growing seasons. Flowering, véraison, and harvest timings of trehalose-treated grapevines were comparable to those of non-treated (hereinafter, control) grapevines. No phenological differences in shoots and leaves were observed among the grapevines tested, whereas smaller bunches as a result of trehalose treatment were observable by the naked eye at harvest (Figure 8A and Figure 9A,B). Bunch weight and length of trehalose-injected grapevines were significantly less than those of control grapevines (Figure 8B,C and Figure 9C,D). The number of berries per bunch of trehalose-injected grapevines was decreased by approximately 50% (Figure 8D) and 60% (Figure 9E) relative to that of control grapevines in the 2018 and 2020 growing seasons, respectively. The weight of ten berries from water- or trehalose-treated grapevines was significantly lower than that from control grapevines in the 2018 growing season (Figure 8E), but there was no difference in berry weight among treatments in the 2020 growing season (Figure 9F). Water injection and spraying or trehalose spraying of buds had no effect on bunch weight and length nor on the number of berries per bunch compared with control grapevines. Interestingly, the number of pedicels per inflorescence was comparable among all the treatments (Figure 9G), suggesting that trehalose did not affect the development of pedicel branching on inflorescences.

No differences in soluble solids content of berries nor in anthocyanins in berry skins were observed among the grapevines tested in the 2018 growing season (Figure 10A,C). Total acids contents in berries from trehalose-treated grapevines were significantly higher than those from control grapevines and water-treated grapevines, regardless of the method of trehalose application (Figure 10B).

Taken together, these results suggest that trehalose injection into buds nearing bud break decreased the number of berries per bunch.

## 4. Discussion

SRA has been identified as trehalose 6-phosphate phosphatase [11], which catalyzes the transformation of trehalose-6-phosphate into trehalose [27]. Trehalose 6-phosphate phosphatase alters inflorescence architecture in maize [10], whereas grape SRA negatively regulates rachis development on the lateral branch of grape inflorescences, thereby decreasing the number of flower buds per inflorescence in VvSRA-overexpressing *Arabidopsis* plants [13]. In the present study, VvSRA drastically upregulated gene expression of *AtCKX7*, which is an *Arabidopsis* homolog of OsCKX2. OsCKX2 regulates cytokinin level to control flower number [21]. The reduction of OsCKX2 expression causes cytokinin accumulation in inflorescence meristems and increases the number of flowers. We also showed that trehalose upregulated *VvCKX5* gene expression in grapevines. VvCKX5 overexpression decreased the number of flowers per inflorescence in *Arabidopsis* transgenic plants as well as in *VvSRA*-overexpressing *Arabidopsis* plants [13]. Finally, in the field experiments, trehalose injection into buds nearing bud break decreased the number of berries per bunch. Here, unfortunately, we could not determine the cytokinin and trehalose levels in VvSRA- and VvCKX5-overexpressing *Arabidopsis* plants nor in the grapevine meristems treated with trehalose due to technical problems. The phenotypes of VvCKX5-overexpressing *Arabidopsis* plants are well known in cytokinin-deficient plants [28], and this would be expected when overexpressing a CKX from another species. Taken together with other circumstantial evidence, we predicted that trehalose produced by VvSRA might promote cytokinin degradation through the activation of VvCKX5, thereby causing a decrease in the number of berries per bunch. Future studies employing alternation of plant hormone profiles including cytokinin, indole-3-acetic acid (IAA), and abscisic acid in buds treated with trehalose would reveal the effect of trehalose on plant hormone homeostasis in grapevine floral development.

Is the predicted crosstalk pathway between SRA-mediated trehalose metabolism and CKX-mediated cytokinin degradation for the determination of flower number specific to grapevines? Trehalose metabolism functions as a transcriptional regulator at the upstream of RA1 during inflorescence development and controls inflorescence architecture in maize [11]. In rice, cytokinin oxidase/dehydrogenase OsCKX2 controls flower number [21]. However, so far, we could not find any positive reports suggesting this crosstalk in other plant species. Future studies employing model plants and crops could reveal the relation between trehalose metabolism and cytokinin degradation for determining the number of flowers per inflorescence. VvCKX5 overexpression in *Arabidopsis* plants decreased the number of flowers per inflorescence but had no effect on the number of secondary inflorescences branching from the main inflorescence. Similarly, exogenous trehalose application into swelling buds nearing bud break in field-grown plants did not affect the development of pedicel branching on inflorescences. The phenotype of the VvCKX5-overexpressing plants was the same as that of VvSRA-overexpressing *Arabidopsis* plants [13], suggesting that VvCKX5, as well as VvSRA-mediated trehalose metabolism, regulates flower formation but not inflorescence branching. This observation supports the finding that the abundance of *VvCKX5* transcripts in juvenile grape inflorescences decreased the number of flowers per bunch. The reduction of rice CKX2 increased grain number through cytokinin accumulation in meristems of inflorescences [21]. The double mutation of *CKX3* and *CKX5* in *Arabidopsis* caused the formation of more and larger flowers by delaying cell differentiation in the reproductive meristem [28]. Thus, regardless of whether we are talking about mono- or dicotyledonous plants, plant CKXs negatively regulate the number of flowers per inflorescence through the enzymatic degradation of cytokinin. Here, we propose that grape VvCKX5 is one of the negative regulators of the number of flowers per inflorescence in grapes.

In the present study, we predicted that trehalose metabolism crosstalked with cytokinin degradation by CKXs through trehalose, thereby leading to the suppression of flower development in grape buds. However, so far, we could not demonstrate any results related to the transition of plant hormones in grapevine meristems treated with trehalose. Indole-3-acetic acid (IAA) downregulates the synthesis of cytokinin in bud regulation [29]. IAA and cytokinin biosynthesis is activated during the release of buds from endodormancy in grapevines, promoting bud sprouting [30]. In *Brassica napus*, *CKX* gene expression was quickly and strongly induced in response to IAA [31]. In contrast, *CKX* gene expression was induced by cytokinin and abscisic acid in maize [32]. In the present study, we demonstrated that NAA upregulated *VvCKX5* gene expression in grapevine cultured cells. Auxin derived from the shoot apex might regulate cytokinin levels by inducing cytokinin degradation through the regulation of CKX expression in peas [33]. The interaction of plant hormones regulates developmental programs through crosstalk in which the hormones affect the expression of genes related to hormone metabolism. Thus, plant CKXs might be related to the complex dynamic balance of hormones in plant tissues, thereby regulating plant development and growth.

## 5. Conclusions

In field experiments over two growing seasons, we demonstrated that trehalose injection into buds nearing bud break decreased the number of berries per bunch. Spraying trehalose on the buds did not decrease the number of berries per bunch. As water injection into the buds did not affect the number of berries per bunch, the effect of the injection itself was negated. Since it might be essential to deliver trehalose to grape meristems in the buds to decrease the number of berries per bunch, injecting trehalose into the buds using a microsyringe, for example, is absolutely necessary. Vineyard operators in Japan face the issue of Pinot Noir berry splitting due to heavy rainfall before véraison. Here, we propose adopting trehalose injection as one of the vinicultural practices to loosen tight bunches without altering berry growth and quality. Although 10% trehalose is a high amount and is not unphysiological in grapevines, trehalose is not inexpensive and might be more cost-effective than other techniques. However, injection of trehalose into each bud using a microsyringe does not seem feasible in commercial vineyards due to requiring long labor hours and large labor load. To further explore the applicability of trehalose injection in viticulture, the development of an injection apparatus that is universally usable may contribute to popularizing this labor-saving technique to loosen tight bunches of wine grapes in viticulture.

## Figures and Tables

**Figure 1 cells-09-02378-f001:**
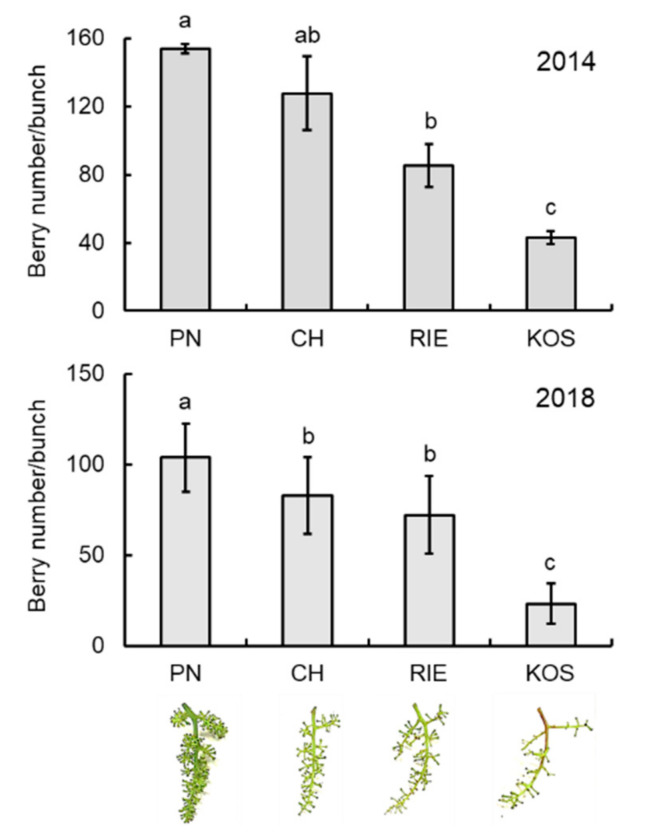
Number of berries per bunch. Bunches were collected in the 2014 and 2018 growing seasons. Bars indicate means ± standard deviations of ten bunches. Different letters above the columns indicate statistically significant difference (*p* < 0.05, Dunnett’s test). PN, Pinot Noir. CH, Chardonnay. RIE, Riesling. KOS, Koshu. Phenotypes of inflorescence are indicated below the graph.

**Figure 2 cells-09-02378-f002:**
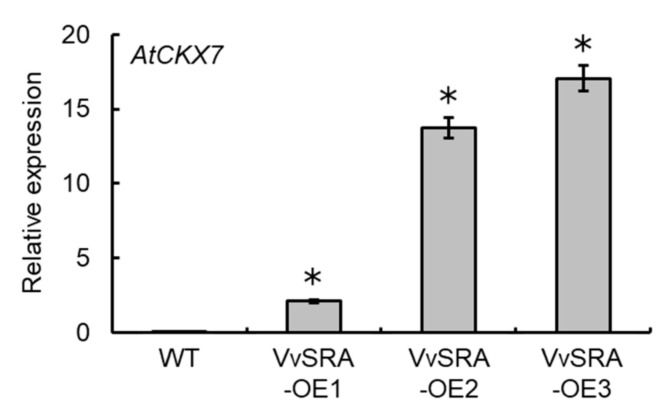
Upregulation of *AtCKX7* in VvSRA-overexpressing plants. Total RNA was isolated from 12-day-old seedlings and subjected to real-time RT-PCR. Actin was used as an internal control. Data were calculated as gene expression level relative to actin gene expression level. Bars indicate means ± standard deviations of three seedlings. * *p* < 0.01 compared with wild type (Dunnett’s test). VvSRA-OE1, -OE2, and -OE3 were VvSRA-overexpressing *Arabidopsis* plants prepared in our previous study [13]. WT, wild type.

**Figure 3 cells-09-02378-f003:**
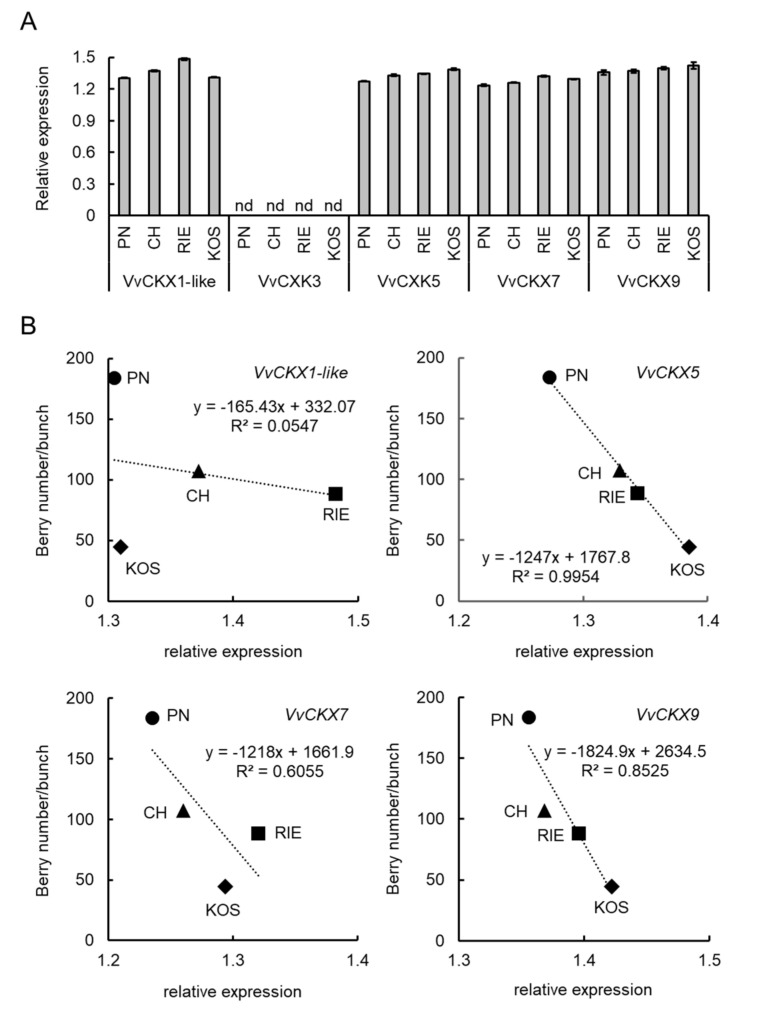
Relationship between number of berries per bunch and *VvCKX* expression in juvenile grape inflorescences. (**A**) *VvCKX* expression levels. Total RNA was isolated from juvenile grape inflorescences (approximately 5–10 mm in the longitudinal axis) of each cultivar and subjected to real-time RT-PCR. Actin was used as an internal control. Data were calculated as gene expression level relative to actin gene expression level. Bars indicate means ± standard deviations of three independent experiments. nd, not detected. (**B**) Regression lines between the number of berries per bunch (obtained from the 2018 growing season in Figure 1) and *VvCKX1-like*, *VvCKX5*, *VvCKX7*, or *VvCKX9* expression in the juvenile inflorescences were drawn, respectively. Scatter plots of the number of berries per bunch versus each gene expression in the juvenile inflorescences were created. Regression lines and R-squared values were calculated from the scatter plots. PN, Pinot Noir. CH, Chardonnay. RIE, Riesling. KOS, Koshu.

**Figure 4 cells-09-02378-f004:**
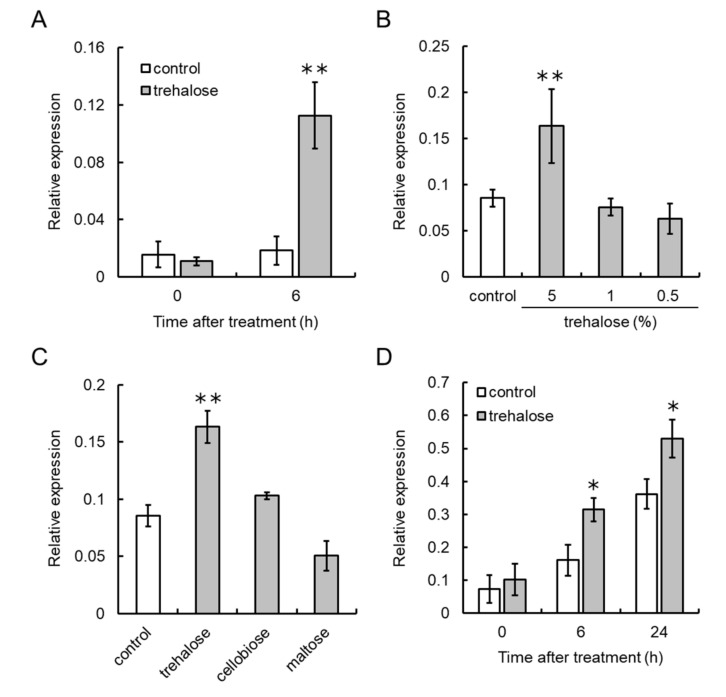
Exogenous trehalose upregulates *VvCKX5* gene expression in VR cells and juvenile grape inflorescences. (**A**) Transcription of *VvCKX5* in VR cells treated with 5% trehalose. (**B**) Transcription of *VvCKX5* in VR cells treated with 0.5, 1%, and 5% trehalose at 6 h post treatment. (**C**) Transcription of *VvCKX5* in VR cells treated with 5% of each disaccharide at 6 h post treatment. (**D**) Transcription of *VvCKX5* in juvenile grape inflorescences treated with 5% trehalose. VR cells and juvenile inflorescences from field-grown grapevines were treated with trehalose or other disaccharides. Total RNA was isolated from VR cells and inflorescences and subjected to real-time RT-PCR. Actin was used as an internal control. Data were calculated as gene expression level relative to actin gene expression level. Bars indicate means ± standard deviations of five independent experiments. * *p* < 0.05 compared with control (Student’s *t*-test for D). ** *p* < 0.01 compared with control (Student’s *t*-test for A or Dunnett’s test for B and C). Control, no treatment.

**Figure 5 cells-09-02378-f005:**
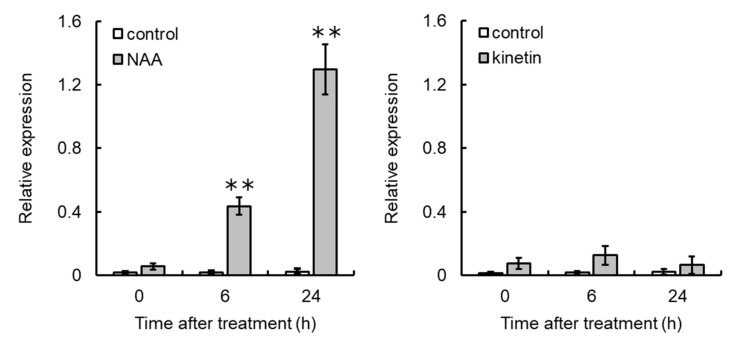
Exogenous NAA upregulated *VvCKX5* gene expression in VR cells. VR cells were treated with excess amounts of NAA or kinetin. Total RNA was isolated from VR cells and subjected to real-time RT-PCR. Actin was used as an internal control. Data were calculated as gene expression level relative to actin gene expression level. Bars indicate means ± standard deviations of five independent experiments. ** *p* < 0.01 compared with control (Student’s *t*-test).

**Figure 6 cells-09-02378-f006:**
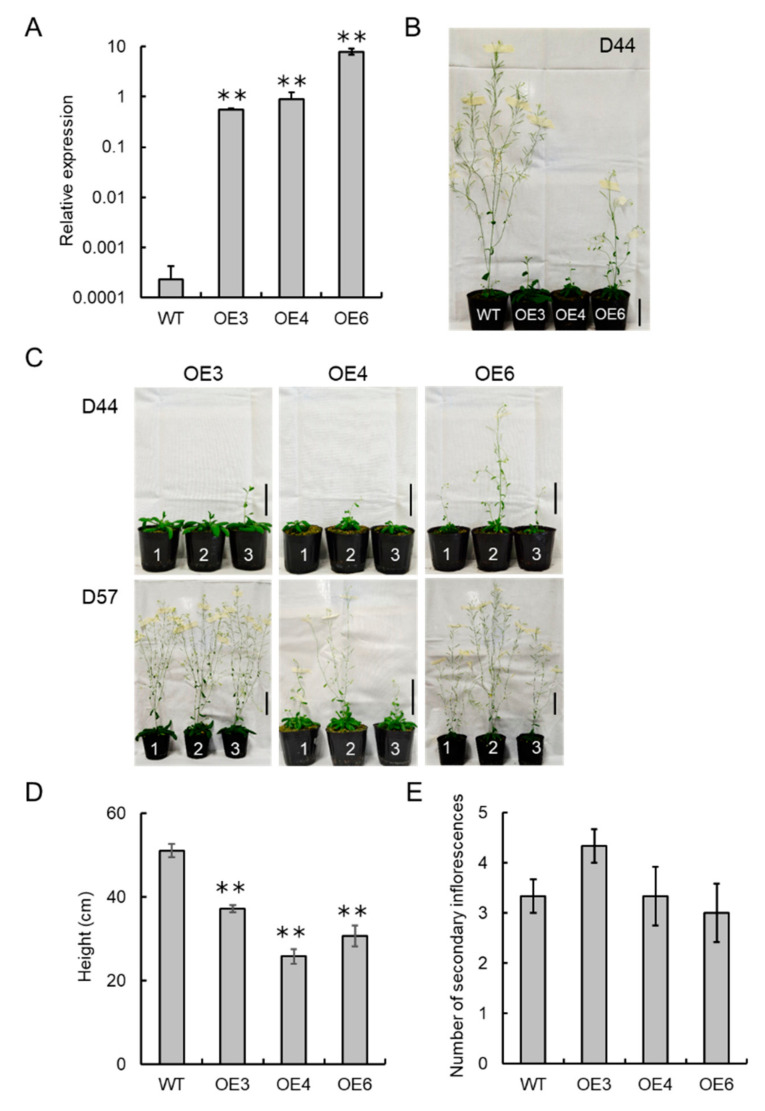
Phenotypes of *Arabidopsis* plants overexpressing VvCKX5. (**A**) *VvCKX5* expression levels in VvCKX5-overexpressing plants. Three independent *Arabidopsis* lines overexpressing VvCKX5 (VvCKX5-OE3, VvCKX5-OE4, and VvCKX5-OE6) were obtained. Total RNA was isolated from rosette leaves of 36-day-old plants and subjected to real-time RT-PCR. Actin was used as an internal control. Data were calculated as gene expression level relative to actin gene expression level. Bars indicate means ± standard deviations of three independent plants. ** *p* < 0.01 as compared with wild type (Dunnett’s test). (**B**) Photograph of transgenic and wild type plants on day 44 after sowing. Scale bar = 5 cm. (**C**) Phenotype of transgenic plants on days 44 (D44) and 57 (D57) after sowing. Three plants were observed for each transgenic plant. Scale bar = 5 cm. (**D**) Plant height. Plant height was measured on day 44 (wild type) or 57 (transgenic) after sowing. Bars indicate means ± standard deviations of three independent plants. ** *p* < 0.01 as compared with wild type (Dunnett’s test). (**E**) Number of secondary inflorescences on wild type and transgenic plants. The number of secondary inflorescences on wild type and transgenic plants was measured on day 42 (wild type) or 57 (transgenic) after sowing, respectively. Bars indicate means ± standard deviations of three independent plants. WT, wild type. OE3, VvCKX5-OE3. OE4, VvCKX5-OE4. OE6, VvCKX5-OE6.

**Figure 7 cells-09-02378-f007:**
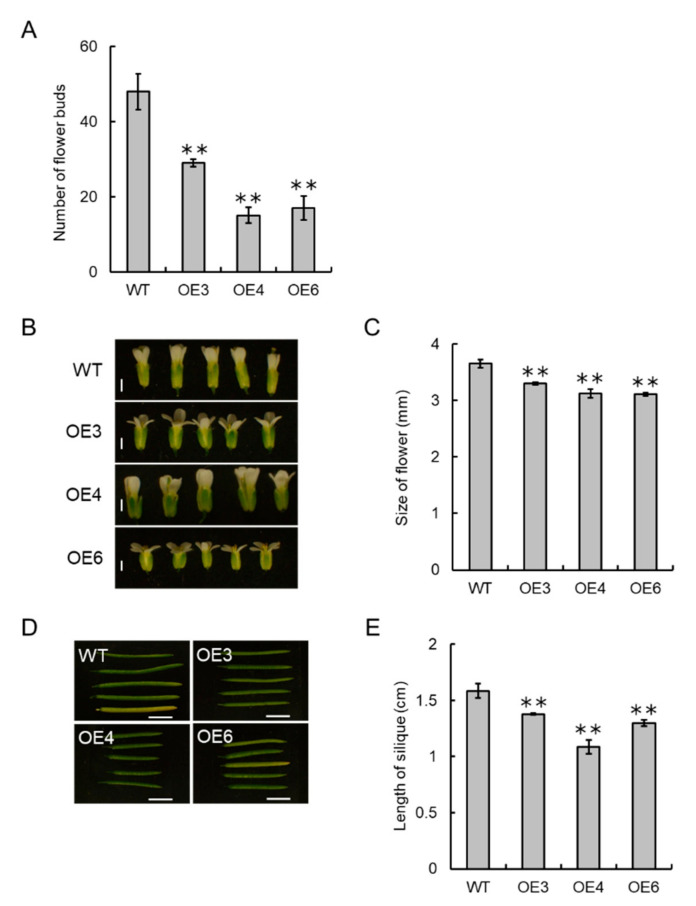
Overexpression of VvCKX5 decreases the number of flower buds per inflorescence in *Arabidopsis* plants. (**A**) Number of flower buds per inflorescence in transgenic and wild type plants. The number of flower buds per inflorescence in each plant was measured on day 42 (wild type) or 57 (transgenic) after sowing. Bars indicate means ± standard deviations of three independent plants. ** *p* < 0.01 as compared with wild type (Dunnett’s test). (**B**) Phenotypes of flowers on day 35 (wild type) or 52 (transgenic) after sowing. Scale bar = 1 mm. (**C**) Flower size. Sizes of flowers on inflorescences of wild and transgenic plants were measured on day 35 (wild type) or 52 (transgenic) after sowing, respectively. Bars indicate means ± standard deviations of five independent flowers. ** *p* < 0.01 as compared with wild type (Dunnett’s test). (**D**) Phenotypes of siliques on day 42 (wild) or 69 (transgenic) after sowing. Scale bar = 5 mm. (**E**) Silique length. Lengths of siliques on inflorescences of wild and transgenic plants were measured on day 42 (wild type) or 69 (transgenic) after sowing, respectively. Bars indicate means ± standard deviations of five independent siliques. ** *p* < 0.01 as compared with wild type (Dunnett’s test). WT, wild type. OE3, VvCKX5-OE3. OE4, VvCKX5-OE4. OE6, VvCKX5-OE6.

**Figure 8 cells-09-02378-f008:**
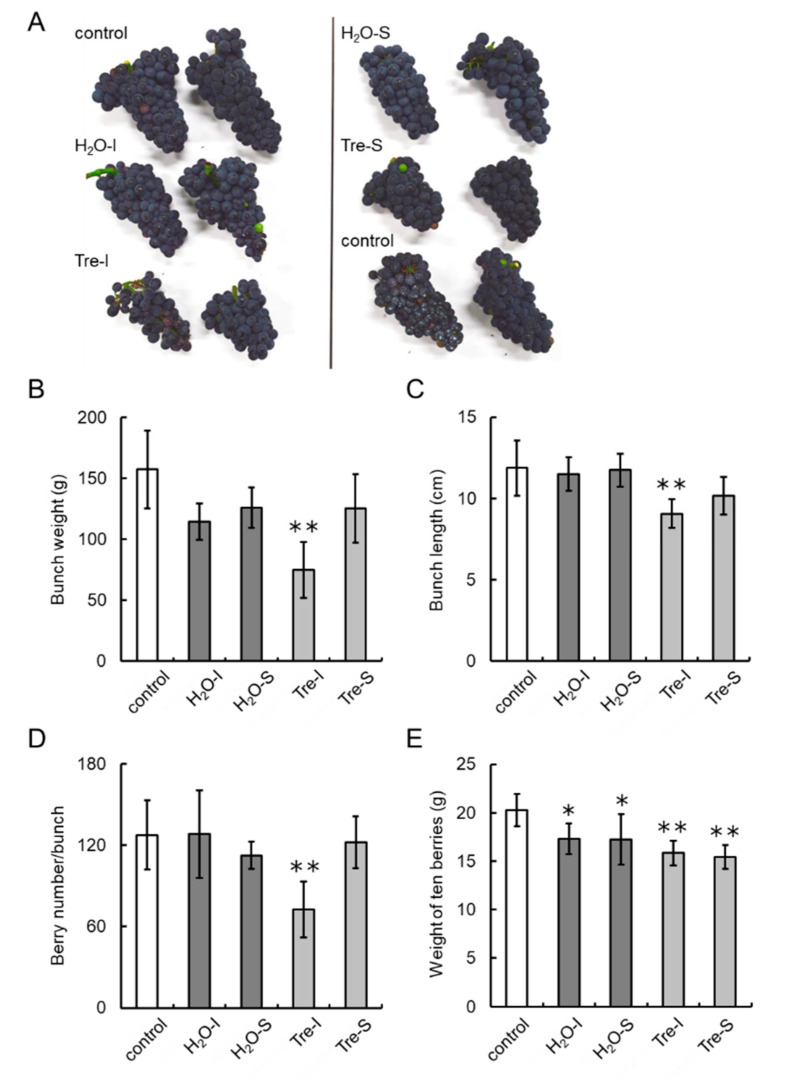
Effects of trehalose injection into buds nearing bud break on bunch characteristics in the 2018 growing season. Ten grapevine buds nearing bud break (Eichhorn–Lorenz Stage 3) were treated with 10% trehalose or water using a microsyringe or an atomizer on 4 April 2018. Young shoots (Eichhorn–Lorenz Stages 5–7) from the buds were again treated with trehalose or water using an atomizer 10 days post first treatment. Twenty bunches from ten shoots (two bunches/shoot) were collected at harvest (Eichhorn–Lorenz Stage 38). (**A**) Photographs of bunches at harvest. (**B**) Bunch weight. (**C**) Bunch length. (**D**) Number of berries per bunch. (**E**) Weight of ten berries. Bars indicate means ± standard deviations of twenty bunches. * *p* < 0.05 compared with control (Dunnett’s test). ** *p* < 0.01 compared with control (Dunnett’s test). Control, non-treated grapevines. H_2_O-I, water-injected grapevines. H_2_O-S, water-sprayed grapevines. Tre-I, trehalose-injected grapevines. Tre-S, trehalose-sprayed grapevines.

**Figure 9 cells-09-02378-f009:**
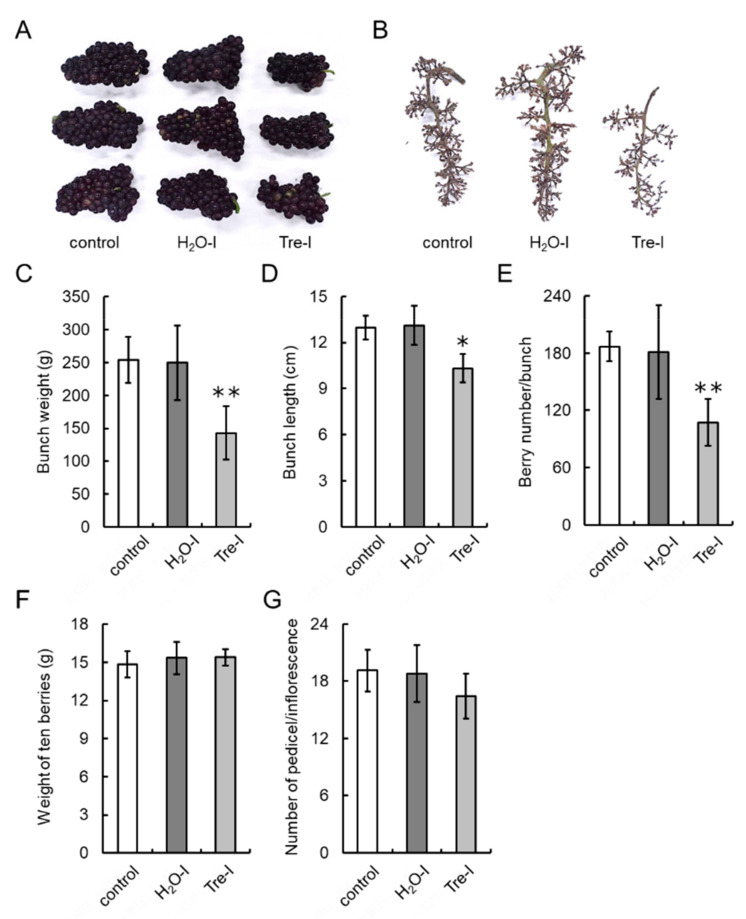
Effects of trehalose injection into buds nearing bud break on bunch characteristics in the 2020 growing season. Ten grapevine buds nearing bud break (Eichhorn–Lorenz Stage 3) were treated with 10% trehalose or water using a microsyringe or an atomizer on 8 April 2020. Young shoots (Eichhorn–Lorenz Stages 5–7) from the buds were again treated with trehalose or water using an atomizer 10 days post first treatment. Twenty bunches from ten shoots (two bunches/shoot) were collected at harvest (Eichhorn–Lorenz Stage 38). (**A**) Photograph of bunches at harvest. (**B**) Photograph of inflorescences. (**C**) Bunch weight. (**D**) Bunch length. (**E**) Number of berries per bunch. (**F**) Weight of ten berries. (**G**) Number of pedicels per inflorescence. Bars indicate means ± standard deviations of twenty bunches. * *p* < 0.05 compared with control (Dunnett’s test). ** *p* < 0.01 compared with control (Dunnett’s test). Control, non-treated grapevines. H2O-I, water-injected grapevines. Tre-I, trehalose-injected grapevines.

**Figure 10 cells-09-02378-f010:**
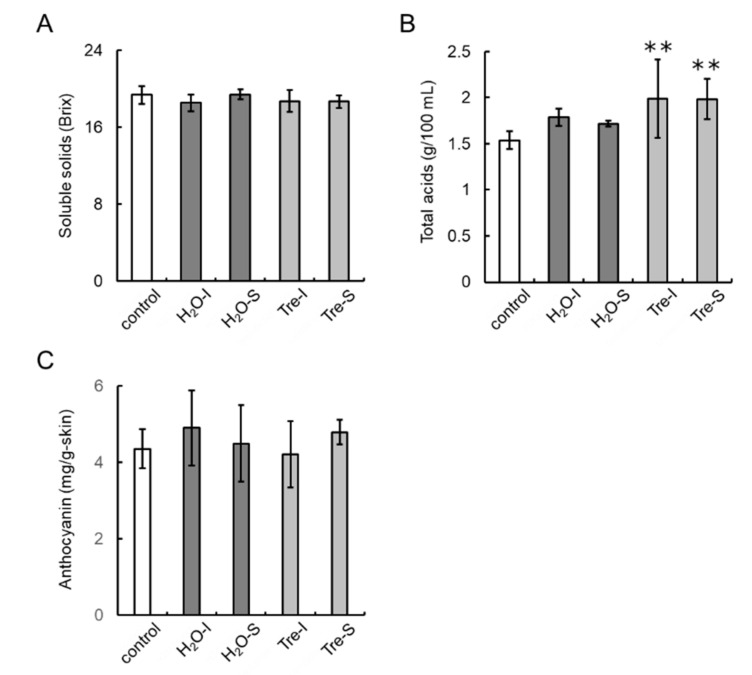
Effects of trehalose injection into buds nearing bud break on berry characteristics. Juices were obtained from each bunch (bunches are shown in Figure 8) by hand-pressing. Skins of ten berries from each bunch were peeled off with tweezers. (**A**) Soluble solids content. (**B**) Total acids content. (**C**) Anthocyanin content. Bars indicate means ± standard deviations of twenty bunches. ** *p* < 0.01 compared with control (Dunnett’s test). Control, non-treated grapevines. H_2_O-I, water-injected grapevines. H_2_O-S, water-sprayed grapevines. Tre-I, trehalose-injected grapevines. Tre-S, trehalose-sprayed grapevines.

**Table 1 cells-09-02378-t001:** Simple linear regression analysis: berry number versus *VvCKX* expression in juvenile inflorescence.

Independent Variable	Partial Regression Coefficient	Standard Error	Standardized Partial Regression Coefficient	Tests of Significance of Partial Regression Coefficient		
				F-Value	*t*-Value	*p*-Value
*VvCKX5* expression	–1247.0	60.242	–0.9977	428.48	–20.7	0.0023
*VvCKX7* expression	–1218.0	695.20	–0.7781	3.0696	–1.752	0.2219
*VvCKX9* expression	–1824.9	536.71	–0.9233	11.562	–3.4	0.0767

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
