# Peer review of "Crosstalk Pathway between Trehalose Metabolism and Cytokinin Degradation for the Determination of the Number of Berries per Bunch in Grapes"

_cells, 2020, doi:10.3390/cells9112378_

Round 1
Reviewer 1 Report
The authors studied the relationship between trehalose metabolism and cytokinin degradation and its impact on grape cluster architecture. However, the CK concentration was not measured in the grape bunch development processes, which is a fundamental element to support the conclusions of the work.
Suggestions
L43. "However, gibberellin application was not carried out in viticulture", please specify this sentence, since in table grape GA application is commonly used it. Maybe, this is correct for wine purposes.
L57. RA1=RAMOSA1?
L58. SRA = RAMOSA3?
L74. In the introduction, more information is needed on cytokinins' role in plant flowering and Vitis sp. It is very forced the presentation of the hypothesis in the text.
L99. Please add the number of days after flowering; the local conditions could be quite different.
L165. Underlined?
L217. Measurement of cytokinin levels is required.
L261. Please show the expression data of CKXs to be cultivated in bar charts (Figure 3 consider it supplementary).
L661. What is the mechanism that links the metabolism of trehalose and CKX?
L664. Please check IAA CK relationships discussed in J Plant Growth Regul 36, 814–823 (2017)
L684. The model presented needs to study the CK balance directly measuring its concentration in each case or its synthesis using an analogous strategy to CKX analyzing Vv IPTs.
L690. Spraying trehalose on the buds did not decrease the berry number on a bunch. Why did it not work? Why did the injection technique affect the phenotype?
L695. It seems not feasible to inject each cluster. Please verify this assumption.
Author Response
Re: revision (Ms. Ref. No.: cells-973630)
Dear Reviewer 1,
Thank you for your decision of October 19, 2020, informing us of the Editorial decision of Cells on our manuscript entitled, “Crosstalk pathway between trehalose metabolism and cytokinin degradation for the determination of grape berry number on a bunch” (Ms. No. : cells-973630). We appreciate very much your constructive criticisms on our manuscript. We would like to submit our revised manuscript according to your comments. Our point-by-point response to the comments and detailing all changes made on the revised manuscript is as follows.
We very much appreciate your helpful suggestions. We hope that our answers are satisfactory and the revised manuscript is now acceptable for publication in the Cells.
Best wishes and thank you very much for your consideration.
Sincerely yours,
Shunji Suzuki
Reviewer’s comments:
- The authors studied the relationship between trehalose metabolism and cytokinin degradation and its impact on grape cluster architecture. However, the CK concentration was not measured in the grape bunch development processes, which is a fundamental element to support the conclusions of the work.
Answer: Thank you very much for your advice to improve our manuscript. We agree that the concentration of cytokinin is an important line of our study, however, we could not determine cytokinin levels in grapevine tissues treated with trehalose due to technical problems.
The phenotypes of VvCKX5-overexpressing Arabidopsis plants we mentioned (Figures 6 and 7) are well known in cytokinin-deficient plants and this would be expected when over-expressing a CKX from another species (Bartrina et al. 2011. Plant Cell 23:69-80). Considering all of data we showed in our manuscript, we think that it is valid to predict that trehalose produced by VvSRA might promote cytokinin degradation through the activation of VvCKX5.
We have now acknowledged this and suggested it as a topic for further research in the Discussion section of the revised manuscript as follows:
‘Here, although unfortunately, we could not determine cytokinin and trehalose levels in VvSRA- and VvCKX5-overexpressing Arabidopsis plants and in grapevine meristems treated with trehalose due to technical problems., The phenotypes of VvCKX5-overexpressing Arabidopsis plants were well known in cytokinin-deficient plants [28] and this would be expected when over-expressing a CKX from another species. Taken together with other circumstantial evidence, we predicted that trehalose produced by VvSRA might promote cytokinin degradation through the activation of VvCKX5, thereby causing the decrease in berry number on a bunch. Future studies employing alternation of plant hormone profiles including cytokinin, IAA, and abscisic acid in buds treated with trehalose would reveal the effect of trehalose on plant hormone homeostasis in grapevine floral development.’
(See p. 13-14, lines 650-658, please)
- L43. "However, gibberellin application was not carried out in viticulture", please specify this sentence, since in table grape GA application is commonly used it. Maybe, this is correct for wine purposes.
Answer: We agreed with the reviewer’s suggestion, and revised the sentence as follows:
‘However, gibberellin application was not carried out in viticulture for wine grapes.’
(See p. 1, line 43, please)
- L57. RA1=RAMOSA1?
Answer: Yes, it is. The abbreviation (RA1) was defined in parentheses the first time in the revised manuscript.
(See p. 2, line 55, please)
- L58. SRA = RAMOSA3?
Answer: No, it isn’t. SRA is SISTER OF RAMOSA3. The abbreviation (SRA) has been defined in parentheses the first time in the revised manuscript.
(See p. 2, line 56, please)
- L74. In the introduction, more information is needed on cytokinins' role in plant flowering and Vitis sp. It is very forced the presentation of the hypothesis in the text.
Answer: We have taken the reviewer’s suggestion, and added the description about the relationship between cytokinin and plant flowering as follows:
‘In the present study, we propose the hypothesis that VvSRA-mediated trehalose metabolism activates cytokinin oxidase/dehydrogenase (CKX)-mediated cytokinin degradation in meristems of inflorescence, thereby decreasing the number of berries on a bunch. CKX catalyzes irreversible cytokinin degradation [18]. Cytokinin promotes floral transition in Arabidopsis plant [19]. In grapevine, cytokinin is implicated in the control of inflorescence formation, differentiation of flowers, pistil development, and berry development [20]. However, the role of cytokinin in the control of berry number on a bunch of each cultivar is not well understood. In rice, the reduction of cytokinin oxidase/dehydrogenase (OsCKX2) gene expression causes cytokinin accumulation in meristems of inflorescence and results in an increase of grain number in main panicle [21].’
(See p. 2, lines 78-86, please)
Then, two additional references [19 and 20] were listed in the revised reference list.
(See the revised Reference list, please)
- L99. Please add the number of days after flowering; the local conditions could be quite different.
Answer: We added the information in the revised manuscript.
(See p. 3, line 107, please)
- L165. Underlined?
Answer: Underlines were added on the nucleotide sequences of PCR primers.
(See p. 4, line 170 and 171, please)
- L217. Measurement of cytokinin levels is required.
Answer: Same answer as comment #1 of reviewer #1.
- L261. Please show the expression data of CKXs to be cultivated in bar charts (Figure 3 consider it supplementary).
Answer: We added the expression levels of VvCKX in juvenile grape inflorescence in the revised Figure 3A. Also, the legend of Figure 3 was revised.
(See the revised Figure 3, please)
- L661. What is the mechanism that links the metabolism of trehalose and CKX?
Answer: VvSRA might be identified as trehalose 6-phosphate phosphatase that catalyzes the transformation of trehalose-6-phosphate into trehalose. Upregulation of VvSRA might accumulate more trehalose in grapevine meristems. Trehalose upregulated VvCKX5 gene expression in grapevine meristems. Since VvCKX5 might regulate cytokinin level by decomposing cytokinin in grapevine meristems, the number of flowers might be decreased. We demonstrated this predicted mechanism that links the metabolism of trehalose and cytokinin decomposition by VvCKX5 using VvSRA-overexpressing and VvCKX5-overexpressing Arabidopsis plants and trehalose application in the field experiments.
- L664. Please check IAA CK relationships discussed in J Plant Growth Regul 36, 814–823 (2017)
Answer: Thank you for excellent suggestion. We added new sentence about activation of IAA/cytokinin biosynthesis in grapevine buds during break dormancy according to Noriega and Pérez’s paper that you suggested as follows:
‘Indole-3-acetic acid (IAA) downregulates the synthesis of cytokinin in bud regulation [29]. IAA and cytokinin biosynthesis is activated during the release of buds from endodormancy in grapevine, promoting bud sprouting [30].’
(See p. 14, lines 684-687, please)
Then, Noriega and Pérez’s paper [30] was listed in the revised reference list.
(See the revised Reference list, please)
- L684. The model presented needs to study the CK balance directly measuring its concentration in each case or its synthesis using an analogous strategy to CKX analyzing Vv IPTs.
Answer: As the reviewer pointed out, we have to measure cytokinin levels in trehalose-treated grapevine meristems and/or juvenile inflorescence to propose the predicted model in Figure 11. However, unfortunately, we could not determine cytokinin levels in grapevine tissues treated with trehalose due to technical problems and seasonal issues as described above. Therefore, we are afraid that the predicted model (Figure 11) was deleted in the revised manuscript.
- L690. Spraying trehalose on the buds did not decrease the berry number on a bunch. Why did it not work? Why did the injection technique affect the phenotype?
Answer: We guess that it is important that trehalose is delivered to grape meristem in buds. Spraying could not deliver trehalose grape meristem in buds.
We added the description in the revised manuscript as follows:
‘Since it might be essential to deliver trehalose to grape meristem in the buds to decrease berry number on a bunch, injecting trehalose into the buds using a microsyringe, for example, is absolutely necessary.’
(See p. 14, lines 701-702, please)
- L695. It seems not feasible to inject each cluster. Please verify this assumption.
Answer: We have taken the reviewer’s suggestion and revised the sentences as follows:
‘Here, we propose adopting trehalose injection as one of the vinicultural practices to loosen tight bunches without altering berry growth and quality. Although 10% trehalose was high and not unphysiological in grapevine, trehalose is not inexpensive and might be more cost-advantageous than other techniques. However, so far, trehalose injection into each buds using a microsyringe can not reduce labor hours and load and seems not feasible in commercial vineyards. To explore further the applicability of trehalose injection in viticulture, the development of an injection apparatus that is universally usable may contribute to popularizing this labor-saving technique to loosen tight bunches in viticulture for wine grapes.’
(See p. 15, lines 704-713, please)
Reviewer 2 Report
The authors present a work on the role of trehalose metabolism and cytokinin degradation in determining the number of berries on the bunch. The authors identify two key genes in this process and also hypothesize a mechanism of crosstalk between the two metabolic pathways. Interestingly authors also propose a possible applicativeimplication of the proposed results.
The work is well presented and the results are interesting and complete.
Some required revisions are as follows:
- Introduction: As the authors report functional evidence on arabidopsis, additional information on how similar the two species are in regulating this phenotype would be useful.
- Methods and Results: Par. 3.1 and 2.2: phenotipic evaluation of berry number on bunch among cultivars was only reported for 2018 season, but any phenotypic trait should be evaluated at least for two seasons. Please provide data for another season. Moreover, in material and method section 2.2, authors report “five bunches were collected…”, while in results, Figure 1 it was reported "ten bunches". Please indicate correct procedure.
- Results:
- par 3.2: please provide also data for the actin gene used as control.
- par 3.3: five genes coding for CKX were selected and analyzed, but data on VvCKX1-like were not commented: Figure 3 showed that linear model does not fit with expression level observed due to the PN and KOS results; please provide comment also about this gene. Moreover, author should provide for all the five genes also expression values other than regression analysis.
- Discussion: authors present an interesting model of a predicted crosstalk pathway between VvSRA-mediated threalose metabolism and cytokine degradation, however authors should provide a better description of novelties of the present work with respect to literature data and differences between vitis and other model species.
- Conclusion: The possible application of trehalose in the field is potentially very interesting, but the authors should better describe what are the current strategies not only in Japan and what are the advantages of the proposed strategy compared to the state of the art.
Author Response
Re: revision (Ms. Ref. No.: cells-973630)
Dear Reviewer 2,
Thank you for your decision of October 19, 2020, informing us of the Editorial decision of Cells on our manuscript entitled, “Crosstalk pathway between trehalose metabolism and cytokinin degradation for the determination of grape berry number on a bunch” (Ms. No. : cells-973630). We appreciate very much your constructive criticisms on our manuscript. We would like to submit our revised manuscript according to your comments. Our point-by-point response to the comments and detailing all changes made on the revised manuscript is as follows.
We very much appreciate your helpful suggestions. We hope that our answers are satisfactory and the revised manuscript is now acceptable for publication in the Cells.
Best wishes and thank you very much for your consideration.
Sincerely yours,
Shunji Suzuki
Reviewers' comments:
- The authors present a work on the role of trehalose metabolism and cytokinin degradation in determining the number of berries on the bunch. The authors identify two key genes in this process and also hypothesize a mechanism of crosstalk between the two metabolic pathways. Interestingly authors also propose a possible applicative implication of the proposed results. The work is well presented and the results are interesting and complete.
Answer: Thank you very much for your advice to improve our manuscript. The manuscript was revised according to the reviewers’ suggestions. We hope that our revision is satisfactory and the revised manuscript is now acceptable for publication in the Cells.
- Introduction: As the authors report functional evidence on arabidopsis, additional information on how similar the two species are in regulating this phenotype would be useful.
Answer: We have taken the reviewer’s suggestion, and revised the description as follows:
‘In grapevine, the transcription level of V. vinifera SRA, VvSRA, is negatively correlated with rachis development on the lateral branch of V. vinifera inflorescence, thereby decreasing berry number on a bunch in cultivars expressing VvSRA abundantly [13]. Concomitantly, the overexpression of VvSRA in Arabidopsis plants drastically decreased the number of flower buds on secondary inflorescence [13]. In addition, VvSRA overexpression induced smaller flowers, shorter inflorescence, and expanded rosette leaves in Arabidopsis plants [13].’
(See p. 2, lines 58-64, please)
- Methods and Results: Par. 3.1 and 2.2: phenotipic evaluation of berry number on bunch among cultivars was only reported for 2018 season, but any phenotypic trait should be evaluated at least for two seasons. Please provide data for another season.
Answer: We have taken the reviewer’s suggestion, and added phenotypic data collected in another season (2014 growing season) in Figure 1.
(See the revised Figure 1, please)
- Methods and Results: Moreover, in material and method section 2.2, authors report “five bunches were collected…”, while in results, Figure 1 it was reported "ten bunches". Please indicate correct procedure.
Answer: ‘Ten bunches’ is correct. We revised it in the revised manuscript.
(See p. 3, line 106, please)
- Results: par 3.2: please provide also data for the actin gene used as control.
Answer: In the present study, actin was used for normalization of VvCKX gene expression and the expression levels of VvCKX genes were expressed as a relative value. This means that the relative value of actin is ‘1’ in every sample. Therefore, we think that the expression levels of actin gene are not necessary in the revised manuscript.
- Results: par 3.3: five genes coding for CKX were selected and analyzed, but data on VvCKX1-like were not commented: Figure 3 showed that linear model does not fit with expression level observed due to the PN and KOS results; please provide comment also about this gene. Moreover, author should provide for all the five genes also expression values other than regression analysis.
Answer: Descriptions about VvCKX1-like expression were added in the paragraph.
(See p. 7, line 323 and 326, please)
Also, expression data of VvCKX genes tested in the present study was included in the revised Figure 3A. We revised the legend of Figure 3.
(See the revised Figure 3, please)
- Discussion: authors present an interesting model of a predicted crosstalk pathway between VvSRA-mediated threalose metabolism and cytokine degradation, however authors should provide a better description of novelties of the present work with respect to literature data and differences between vitis and other model species.
Answer: We have taken your suggestion, and added the descriptions in the revised Discussion section as follows:
‘Is the predicted crosstalk pathway between SRA-mediated trehalose metabolism and CKX-mediated cytokinin degradation for the determination of flower number specific to grapevine? Trehalose metabolism function as a transcriptional regulator at the upstream of RA1 during inflorescence development and controls inflorescence architecture in maize [11]. In rice, cytokinin oxidase/dehydrogenase OsCKX2 controls flower number [19]. However, so far, we could not find any positive reports suggesting the crosstalk in other plant species. Future studies employing model plants and crops would reveal the relation between trehalose metabolism and cytokinin degradation for determining flower number on an inflorescence.’
(See p. 14, lines 659-666, please)
- Conclusion: The possible application of trehalose in the field is potentially very interesting, but the authors should better describe what are the current strategies not only in Japan and what are the advantages of the proposed strategy compared to the state of the art.
Answer: We have taken the reviewer’s suggestion and revised the sentences as follows:
‘Here, we propose adopting trehalose injection as one of the vinicultural practices to loosen tight bunches without altering berry growth and quality. Although 10% trehalose was high and not unphysiological in grapevine, trehalose is not inexpensive and might be more cost-advantageous than other techniques. However, so far, trehalose injection into each buds using a microsyringe can not reduce labor hours and load and seems not feasible in commercial vineyards. To explore further the applicability of trehalose injection in viticulture, the development of an injection apparatus that is universally usable may contribute to popularizing this labor-saving technique to loosen tight bunches in viticulture for wine grapes.’
(See p. 15, lines 704-713, please)
Round 2
Reviewer 1 Report
No comments for the authors